# Aging and Family Relationships among Aymara, Mapuche and Non-Indigenous People: Exploring How Social Support, Family Functioning, and Self-Perceived Health Are Related to Quality of Life

**DOI:** 10.3390/ijerph19159247

**Published:** 2022-07-28

**Authors:** Lorena Patricia Gallardo-Peralta, Esteban Sanchez-Moreno, Soledad Herrera

**Affiliations:** 1Dirección de Investigación, Postgrado y Transferencia Tecnológica, Universidad de Tarapacá, Arica 1000000, Chile; 2Department of Social Work and Social Services, Universidad Complutense de Madrid, 28223 Madrid, Spain; 3Institute for Research in Development and Cooperation (IUDC-UCM), Universidad Complutense de Madrid, 28015 Madrid, Spain; esteban.sanchez@cps.ucm.es; 4Institute of Sociology, Pontifical Catholic University of Chile, Santiago 7820436, Chile; mherrepo@uc.cl

**Keywords:** older persons, quality of life, social support, family functioning, self-rated health

## Abstract

Family relationships play a central role in wellbeing among older adults in Chile. Based on the theory of social production functions, this study examined the relationship between perceived social support from children, partners and relatives, family functioning, self-perceived health and quality of life (QoL) among Chilean older adults. The study used a multi-ethnic sample of Chilean older adults living in rural areas in the regions of Arica and Parinacota (north) and Araucanía (south). A model was analyzed that emphasizes relationships differentiated by the source of support, family functioning and self-perceived health in the explanation of QoL. The results obtained from the structural equation modelling (SEM) analysis showed the existence of indirect relationships of social support from children, partners and other family members via family functioning, while self-perceived health was directly associated with QoL. The findings indicate that family functioning is a main variable in the contrasted model, in addition to confirming the importance of distinguishing the role of the various sources of support. Research is needed to examine in detail intergenerational relationships and other relationships with family members who are significant in the wellbeing of older adults. This research corroborates that family relationships have a specificity that needs to be addressed in gerontological social intervention, as well as continuing along the lines of strengthening or improving existing family ties (more quality) over the quantity of social relationships.

## 1. Introduction

As with many other developing countries, Chile has experienced a process of rapid population aging that has not been accompanied by the development of a robust social protection system to secure wellbeing in old age [1]. The Chilean social protection system is weak in terms of its provision of healthcare, a pension system and other guarantees required to enable people to live with dignity, giving rise to large social inequalities among older adults [2,3]. In this context, the family has the main responsibility of caring for, supporting and protecting older adults [4,5]. Intergenerational relationships play a significant role in Chile as a result of the limited coverage offered by the social protection system [6] as well as the traditional multigeneration culture of care, preferably among Latin American indigenous older people [7]. Together with family, other informal social networks such as friends and community play a leading role in generating support and responding to the psychosocial needs of the older age group [8,9].

There has been extensive research into the positive effects of social support from informal networks on physical health [10], mental health [11], psychological wellbeing [12], life satisfaction [13] and quality of life (QoL) [8,14] among Chilean older adults. This study contributes to the existing literature by analyzing the different roles of the sources of family support (from a partner, children, or other family members), taking into account the potentially mediating role of family functioning and self-perceived health in QoL for Chilean older adults. Additionally, given the marked gender gaps in Chilean older adults [5], we seek to identify possible differences between men and women in the analysis of QoL.

### 1.1. Social Support Networks and Quality of Life

Research into the aging process has produced extensive empirical evidence regarding the factors and determinants that drive wellbeing among older adults, with QoL being one of the most commonly analyzed constructs on an international scale [15,16,17]. These studies have examined different dimensions of QoL, including autonomy, functional capacity, intimacy, activities and participation [18]. Intimacy and social integration have a specific value for QoL in old age [19]. The available evidence suggests that access to social support networks contributes to wellbeing by helping older adults to properly adapt to and cope with the various difficulties that are inherent to old age [20,21].

Following this line of argument, the social integration of older adults entails access to functional social support [22], meaning links and interactions that can provide affection, care and a sense of being valued (emotional social support), solve problems or address needs (instrumental social support) or provide advice, information or guidance (informational social support). Moreover, social integration in support networks, particularly involving the family, also involves a sense that one is part of a network of communication and mutual obligation [23]. This multidirectional relationship implies a principle of reciprocity of social support, and the available evidence does suggest that older adults are frequent providers of support for the members of their intimate or close networks [6,24]. Social support hence acts as a form of coping assistance, or the active participation of significant others in an individual’s stress-management efforts [25].

Having access to adequate levels of social support can have a direct effect on QoL for older adults, as well as playing a role in alleviating the potential negative consequences of adverse circumstances that frequently arise during this stage of life, including the loss of significant others, feelings of depression or loneliness, facing illness and other negative experiences [26,27]. Social support is hence directly linked to wellbeing during the aging process [28,29], particularly when the sources of that support are members of close family (a partner and children), but also when the support comes from other members of the extended family including siblings, nieces and nephews and grandchildren [30]. This argument is particularly important in predominantly family-based cultural contexts such as Chile [8].

### 1.2. Hierarchy in Terms of the Provision of Support, Family Functioning and QoL

Family social support can be supplemented by other family members and even replaced in certain situations. Following the classical approach of the hierarchical-compensatory model of social support [31], older adults tend to have an order of preference when choosing their support group, regardless of the type of help that is needed. The hierarchical order tends to be: daughters, sons, family members, friends and more distant neighbors, in descending order. Peters et al. [32] later observed that regardless of whether assistance was instrumental or affective, older adults primarily resorted to their spouse, followed by adult children, friends, siblings and other relatives, and when lacking one of these intimate categories, no compensatory principle operated.

However, for these sources to act as an effective source of social support, the family network needs to be able to provide an adequate, flexible and timely response to the emotional and psychosocial needs of older adults. In other words, the family needs to be functional, with members who perceive it as cohesive and offering the nurturement and resources that are necessary for personal growth and sustenance in the face of life’s challenges [33]. According to Pérez Peñaranda et al. [34], a functional family mobilizes in difficult situations, such as illness or disability, with all of its members participating, supporting and contributing on equal terms to secure collective wellbeing.

Family support networks and their functioning are put to the test in contexts involving meeting the daily needs of older adults. The existing literature shows that family networks play a central role in caring for older adults [35]. The relevance of family networks takes on greater importance in countries with insufficient social security systems. This is the case in Chile, with social protection aimed at subsidizing part of the costs in areas such as health. As a consequence, older people must allocate a significant part of their pensions to expenses associated with the treatment of their illnesses or health problems [4,5]. Thus, when it is necessary to deal with health problems, the family generally has to mobilize its own resources or tools to respond to the needs that have arisen. These problems occur frequently during the aging process and generate a need for family members to mobilize so there are sufficient levels of social support to meet the needs of their older adult members [36,37]. 

### 1.3. Literature Review and Hypotheses

As established in the theory of social production functions (SPF theory) [38], people seek to satisfy their physical and social (affection, behavioral confirmation and status) needs by using available resources and managing resource shortfalls and limitations as well as they can [39]. This approach indicates the importance of analyzing the differentiated role of social support according to its source: a partner, children and other family members. In the case of older people who do not have this network, either because they do not have a partner, children, etc., they will try to mobilize other supplementary family networks. Along these lines, the set of hypotheses formulated for the present study included as a fundamental element the distinction of the sources of family support of the participants, as shown in Figure 1. 

The model shown in Figure 1 constitutes the range of inter-related hypotheses of this study. Differences in the fit of the model were expected to be observed between men and women. The basis for each hypothesis subset is described below:

**H1.** 
*Social support from children is positively associated with QoL.*


**H1a.** 
*Social support from children is positively and directly associated with QoL.*


**H1b.** 
*Social support from children is positively and indirectly associated with QoL, through family functioning.*


The benefits of support from children are closely associated with the satisfaction of psychological needs for connectedness, engaging activity, meaning, security and control and experiencing a positive self, the fulfillment of which appears to be highly correlated to subjective wellbeing [40]. In addition, the assumption of mandatory respect and filial loyalty means that children act as a resource that promotes the physical and social wellbeing of older adults [41], particularly when physical proximity permits high levels of interaction and participation in their daily lives [42]. Margolis and Myrskylä [43] observed, in a study conducted in several countries including Latin America, that the association between number of children and happiness becomes positive and is strongest among those who are likely to benefit most from support from children in their later years. Research conducted by Bélanger et al. [29] confirmed that Latin American older adults who perceived higher levels of social support from children obtained better results in terms of QoL. Later, Gyasi et al. [26] confirmed that the social support children provide to their older adult parents in Ghana can reduce stress and improve health and self-efficacy, all of which has a positive impact on psychological wellbeing. In our study, we adopted the hypothesis that social support from children has a direct and positive relationship with QoL. Support from children was also expected to be indirectly associated with QoL through family functioning.

**H2.** 
*Social support from a partner is positively and indirectly associated with QoL through family functioning.*


Relationships with partners are defined by a context of intimacy and based on commitment, emotional interdependence and affective reciprocity, meaning that partners play a central role in satisfying psychosocial needs in old age [42]. A study conducted by Ryan et al. [44] in the United States showed that perceived social support from a partner was associated with better self-assessed health and fewer functional limitations. Social support from a partner also positively affected QoL [29]. Our study hypothesized that social support from a partner would be associated with QoL not directly, but rather indirectly through family functioning. Previous studies suggest that support from a partner tends to be taken as a given, meaning that a positive influence on QoL should only be expected if the relationship with the partner involves effective and positive interactions [45,46].

**H3.** 
*Social support from other relatives (grandchildren, siblings, nephews and in-laws) is positively and indirectly (through family functioning) associated with QoL.*


Relatives form a significant support network in old age. Social support from this source is particularly important in terms of offering compensation and adaptation when the nuclear family is unable to satisfy social needs [39]. In this regard, new family structures [47] show that older adults do not always have access to social support from a partner or children, whether due to their absence or a lack of availability, in response to which other family members provide social support. Sener et al. [30] reported that life satisfaction among older Turkish women was significantly related to the frequency of contact with siblings and friends. Meanwhile, a study by Harada et al. [48] confirmed that negative interactions with family members living together with an older adult in Japan were associated with worse mental health (distress). A recent study by Stocker et al. [49] in rural Midwestern United States showed that sibling relationships have a positive effect on older adults in terms of feelings of loneliness and of wellbeing (alleviating symptoms of depression, anxiety and hostility); they are effectively the longest-lasting relationships available to most people. As with spouses, however, support from the extended family can only be expected to be positively associated with QoL if the support is related to an improvement in family functioning; in other words, only if the family network facilitates an adequate and timely response to meet the emotional and psychosocial needs of older adults.

**H4.** 
*Family functioning will have a positive and direct association with QoL.*


**H4a.** 
*Family functioning will have a positive and direct association with self-perceived health.*


Several studies have shown that family functioning has a positive impact on the wellbeing of older adults. In the context of mental health, a study conducted by Wang and Zhao [50] concerning depressive symptoms among Chinese older adults according to household structure (with or without an “empty nest”) confirmed the importance of the family environment and its functioning. In the context of physical health, research published by Dumitrache et al. [20] showed that among Spanish older adults with health problems, life satisfaction was related to the size of the family network and satisfaction with family life. It has even been confirmed that having social support and good family functioning is associated with a reduced perception of age-based discrimination [51]. Family functioning was hence expected to be observed to be positively and directly associated with QoL.

Studies mainly report that family functioning is related to health in general and not specifically to self-perceived health. In this regard, a study involving Portuguese older adults confirmed a statistically significant relationship between family functioning and the presence of chronic illnesses, whereby 54.1% of older adults with chronic illnesses classified their family as severely dysfunctional [52]. A later study by Wang et al. [53] with Chinese nonagenarians and centenarians reported that participants with cognitive impairment had lower scores for family functioning than those without cognitive impairment. We therefore expected to observe a direct and positive association between family functioning and self-perceived health in this study.

**H5.** 
*Self-perceived health is directly and positively associated with QoL.*


**H5a.** 
*Self-perceived health is directly and negatively associated with social support from family members.*


Health is a basic component of QoL [18]; for this reason self-perceived health can be expected to have a direct impact on the general assessment of wellbeing in older adults. Self-perceived health is considered one of the most valuable and reliable health assessment indicators [54,55]. A study by Trentini et al. [56] reported a relationship between the subjective perception of one’s state of health and QoL. In our study, we expected to observe the existence of a negative association between self-perceived health and social support from family members, based on the assumption that all members of an older person’s network will mobilize in situations involving health problems, including networks made up of other family members such as grandchildren, siblings, nephews and in-laws.

The hypotheses did not differentiate by ethnic group, although in this study a large number of participants were indigenous. In this sense, the organization of indigenous communities in Chile, Aymara and Mapuche, has a strong family and community component. Rural and indigenous areas are characterized by ageing, which was investigated in this study. This demographic change is due to the migration of the young population to the cities [57]. Even so, the organization of care is articulated from the family and involves the community, i.e., neighbors [58]. Another characteristic feature is the gaps in the health of the indigenous population; in general, indigenous people have worse physical and mental health indicators and problems of access and coverage in health, highlighting a lower life expectancy [59]. In the specific case of older Aymara women, they have more depressive symptoms [60].

Differences between men and women were expected in terms of the fit of the proposed model to the data. In general, the empirical evidence confirms that women have better social support networks in old age in terms of both quantity and quality of links, as well as tending to have more diverse networks [61,62,63]. We hence expected to observe differences in the relationships between the type of support network and QoL. However, the role of family functioning was expected to be the same for men and women: regardless of gender, older adults were expected to perceive greater wellbeing when assessing their family networks as providing a timely and satisfactory response to their emotional and physical needs (feeling that their family is functional). The proposed model was examined using a multigroup analysis for this reason.

## 2. Method

### 2.1. Participants

The study was based on a national and cross-sectional study entitled: “Ageing in context: the influence of the residential environmental and ethnic belonging on successful ageing among older Aymara and Mapuche Chilean adults”. The sample consisted of 800 older adults living in northern and southern Chile. Inclusion criteria were age (60 years or older), residence (rural setting), health status (no severe cognitive impairment) and voluntary participation in the study. A sample stratified by sex, ethnicity and place of residence (municipal or rural areas) was used to ensure representativeness in each of these territories. 

The fundamental features of the sample are set out in Table 1. The average age was 72.07 years (SD = 7.81) and 49% of participants were women, 54% were married or cohabiting and 25% were living alone. The average household size was 1.59 people (SD = 1.72). In terms of education, 54% of the sample had not completed primary school and only 15% had received any education at high-school level or higher. The fact that 71% of the sample were indigenous (35% Aymara and 65% Mapuche) is particularly noteworthy.

### 2.2. Recruitment

Participants were contacted via two procedures. When possible, the research team made first contact directly and arranged an appointment to conduct the interview. The rural areas had low population density, meaning that making contact with the older adult population was relatively straightforward. Some members of the research team (especially social workers) had enjoyed previous access to some communities from which participants were recruited, which also made it easier for the technical team to obtain access. When first contact entailed greater difficulty, it was made via key social agents, including council personnel (mainly social workers) and key neighborhood and local leaders. These agents carried out an initial selection of participants based on the inclusion criteria. The experience and knowledge of the community of social agents contributed to a recruitment process that facilitated the identification of people with dementia (excluded from eligibility), for example. The interviewer attended the place indicated for the interview, usually in the home of the elderly person and rarely in a municipal office that would allow the interviewer to conduct the interview, that is, a place that offered a space of confidentiality, warmth and quietness.

### 2.3. Procedure

A face-to-face interview method was used to collect the data. The questionnaire—comprising various scales as described in the following section—was read aloud to interviewees. It took approximately 45 min and was administered by qualified social work and psychology professionals. Interviewers learned to administer the questionnaire in a short training workshop; specifically, they received instructions on how to address potential difficulties with understanding questions, for which purpose examples and even the linguistic meaning of some terms were provided. The main language used for the scales was Spanish. 

The Ethics Committee of Tarapacá University and the National Council for Science and Technology of Chile approved and monitored the ethical aspects of the study (monitoring reports N°08-2017, 02-2019 and closing report N°01-2020). All procedures performed in studies involving human participants were in accordance with the 1964 Helsinki declaration and its amendments or comparable ethical standards. The data were processed confidentially and anonymously, having first obtained the informed consent of participants.

### 2.4. Measures

Quality of life. The WHOQOL-OLD [18] is a questionnaire used to assess quality of life specifically among older adults. It consists of a 24-item scale that evaluates the following dimensions: sensory abilities; autonomy; past, present and future activities; social participation; death and dying; and intimacy. A Likert-type scale ranging from 1 to 5 is used, meaning the total score for the scale ranges from 24 to 120 points, with higher scores representing greater QoL. This questionnaire has been validated among the Chilean population [64]. The internal consistency index (Cronbach’s alpha) for the general scale was 0.81.

Social support. The Perceived Social Support Questionnaire (PSSQ) produced by Gracia et al. [22] is a scale made up of nine items evaluating the functional dimensions of emotional support (*To what extent could you freely share and express your feelings with this person?*), advice (*To what extent would you turn to this person for help if you needed advice or useful suggestions to resolve difficulties?*) and assistance (*If you were ill and needed to be taken to a doctor, to what extent would this person help you*?). It also measures reciprocity of support with respect to each source, whether emotional support (*If this person were worried, depressed or had personal or family difficulties, would they come to you?*), advice (*If this person needed advice, to resolve a problem or to make an important decision, would they come to you*?) or assistance (*If this person were ill, needed money or had another problem, would they come to you?*). The PSSQ also offers a total score for functional support and reciprocity of support, as well as the number of components of the network, and separate scores for the different sources of social support. Our study took into account perceived social support from a spouse/partner, children and other family members (grandchildren, siblings, nephews and in-laws). The questionnaire had a pre-filter question for each source of support, e.g., do you have a partner, do you have children, etc., so that participants who did not have the network were included as a response category “does not apply because you do not have the network” (category = 0). For those who had the support network, they could answer: 1 = never, 2 = rarely, 3 = sometimes, 4 = quite often and 5 = almost always. The PSSQ has been validated for Chilean older adults [65]. The internal consistency index (Cronbach’s alpha) was adequate for each of the sources analyzed: spouse/partner (0.91), children (0.93) and other family members (0.94).

Family functioning. The Family APGAR questionnaire [66] is a standardized questionnaire used to measure family functioning. It consists of five items that evaluate an individual’s perception of the support offered by their family: adaptation, partnership, growth, affection and resolve. The final score of the questionnaire ranges from 0 to 10 and the cut-off points are as follows: severely dysfunctional family (0–3 points); family with mild dysfunction (4–6 points) and functional family (7–10). The internal consistency index (Cronbach’s alpha) for the general questionnaire was 0.98.

Self-rated health. Self-rated measures are one of the commonest means of evaluating health status in a summarized manner, their usefulness having been demonstrated in a large number of studies [67]. They represent a valid and reliable measure among people without cognitive impairment, allowing respondents to prioritize and evaluate different aspects of their health and thereby maximizing the measure’s sensitivity to respondent views of health [68]. In our research, self-rating health measures were used on a scale ranging from one to five (1 = very poor, 2 = poor, 3 = fair, 4 = very good, 5 = excellent).

### 2.5. Analysis

The testing of hypotheses started with a descriptive analysis. An attempt was then made to adjust the hypothetical model in Figure 1 to the data using structural equation modeling (SEM) analysis, with the maximum likelihood method, supplementing the analysis with the bootstrapping technique to produce unbiased estimators. Finally, a multigroup analysis was performed to analyze potential differences in the fit of the model adjusted in the previous step between men and women. The fit of the SEM models was evaluated using the following indices: RMSEA (Root Mean Square Error of Approximation), CFI (Comparative Fit Index), AGFI (Adjusted Goodness-of-Fit Index), NFI (Normed Fit Index), and NNFI (Non-Normed Fit Index, also known as TLI). For RMSEA, values below 0.05 indicate a good fit, while values of 0.08 or lower suggest an acceptable fit. For CFI, AGFI, NFI and NNFI, values above 0.95 suggest a good fit, while values above 0.90 imply an acceptable fit [69,70]. The analysis was performed using the IBM-SPSS and IBM-AMOS, V25 programs.

## 3. Results

Table 2 shows the descriptive statistics for the study variables, as well as the Pearson correlation coefficients. The results generally provide preliminary support for the hypotheses. Notably, there is a bivariate association between social support from extended family and social support from children and partners.

An SEM analysis was carried out to establish the fit of the hypothesized model (Figure 1) to our study data. There were adequate values for the goodness-of-fit indicators in general, although a significant number of those indicators did not show values indicative of an adequate fit [χ^2^(df 6) = 31.35; *p* < 0.00); AGFI = 0.95; NFI = 0.94; IFI = 0.95; NNFI = 0.88; CFI = 0.95; RMSEA = 0.073 (0.049, 0.099)]. The model was hence adjusted based on theoretical criteria and the modification indices obtained in the analysis. The resulting model is shown in Figure 2, which includes standardized regression coefficients. The main variation compared to the hypothesized model consists of the existence of association coefficients between social support measures, depending on the source. The adjustment of the model included in Figure 2 is good, as the various indicators show [χ^2^(df 5) = 13.70; *p* = 0.018); AGFI = 0.97; NFI = 0.97; IFI = 0.98; NNFI = 0.95; CFI = 0.98; RMSEA = 0.047 (0.018, 0.077)].

To analyze the gender-based differences involved in the association between social support, family functioning and QoL, a multigroup analysis was performed to confirm the fit of the proposed model for men and women. The results of this analysis demonstrated that the model did not fit the subsample of men, insofar as three association coefficients did not result in significant values: (1) self-rated health and social support from other family members (β = −0.16; *p* = 0.40); (2) family functioning and self-rated health (β = 0.20; *p* = 0.11; and (3) social support from children and social support from other family members (β = 0.08; *p* = 0.07). Taking these results into consideration, the model was simplified by eliminating those associations. The outcome is shown in Figure 3, which comprises a model with an acceptable fit [χ^2^(df 16) = 56.79; *p* < 0.00); AGFI = 0.94; NFI = 0.90; IFI = 0.92; NNFI = 0.85; CFI = 0.92; RMSEA = 0.057 (0.041, 0.073)] and for which it was possible to establish invariance relating to its configuration and structural coefficients (metric invariance), as can be observed in Table 3. In testing for equivalences across groups (in this case, men and women), the hypothesis that a set of parameters (a model) fits for the groups considered was tested. In this vein, the existence of invariance indicates that the same underlying model is statically valid for both men and women subsamples. Therefore, the results shown in Figure 3 constitute a model that allows us to describe the relationship between social support, family functioning, and QoL for both men and women. Taken as a whole, the comparison of the results shown in Figure 2 and Figure 3 suggests that our hypothesized model adequately fit the data, with substantial differences between men and women that are fundamentally related to the role played by the different sources of social support in the model.

## 4. Discussion

This study examined the relationships between social support, family functioning, self-perceived health and QoL. The contrasted theoretical model, based on SPF theory, put to the test social support from different sources (children, a partner, and other family members), family functioning as a mediating variable of those social networks and the direct association of self-perceived health. The results confirmed the appropriateness of this strategy; they are relevant to understanding the functional aspects of family support networks in QoL among older adults. Family functioning is also a lead variable, with confirmation of the role it plays as a mediator between social support from the various family sources and QoL. When a family can offer a timely response that addresses the social and physical needs of its older adult members, social support from children, a partner and other family members is associated with higher QoL. This study also analyzed the fit of the model from a gender perspective, with the finding that when adjusted to reflect men and women separately (Figure 3), the model suggested that family functioning constitutes a central variable in the analysis of QoL for both groups. In contrast, social support from different sources (children, a partner and relatives) has a specific association with QoL according to gender.

The findings of this study confirm that social support from family networks is associated with QoL for older adults [6,8,28]. This association can be explained because social support is made up of links of intimacy and emotional closeness guided by motivations based on obligation, loyalty or love [61]. As stated by Thoits [71], primary groups made up of family members and relatives tend to be intimate and enduring, with relationships considered important and influential throughout the course of a lifetime. This study assessed social support by examining the perception of receiving social and emotional support, advice and assistance in day-to-day and sometimes difficult circumstances. The measurement of social support in this study also included the reciprocity of emotional support, advice and assistance with the social network. In this sense, social support is a reciprocal process involving the exchange of support in networks that are considered intimate or personal [72].

Turning to the study hypotheses, it was corroborated that children have a highly significant impact on the wellbeing of older adults [26,42], with the findings confirming that social support from children is directly and indirectly associated with QoL (H1, H1a, H1b). Support from children hence plays a fundamental role in understanding QoL, as it can contribute directly to wellbeing but also indirectly through family functioning. It should be recalled that the organization of care for older adults in Chile revolves around intergenerational family relationships, whether for cultural reasons [8] or due to the lack of a robust social welfare system [5]. In this regard, a study conducted by Fernández and Herrera [6] demonstrated that solidarity-based family relationships in Chile between parents and children are based on the rule of filial obligation and reciprocity, in addition to observing that geographical proximity increases the likelihood of older adults receiving support. Taking as the starting point a family-focused social structure that promotes relationships of intergenerational solidarity, there is hence a need to foster adequate family functioning. Children must be able to mobilize their resources to effectively respond to their parents’ emotional and physical needs in old age. The SEM model for the sample as a whole (Figure 2) shows that social support from children is related to social support from relatives, which represents another indirect association with QoL. However, this relationship disappears in the multigroup model (Figure 3). Having support from children only entails an increased perception of support from family members for women. In contrast, this relationship is not seen in the case of men. One possible explanation is the fact that women have more intense relationships with their extended family, mainly through their children and their respective grandchildren, while men do not manage to find support in the extended family through their children [73,74].

Social support from a partner was indirectly associated with QoL (H2) through family functioning, but a direct association was not found. The confirmation of this hypothesis suggests that relationships with partners are mediated by cognitive and affective empathy (empathic concern); that is, understanding the other person’s emotions [72], a process directly related to adequate family functioning. Partner social support has been extensively discussed as a determinant of wellbeing under the assumption of invisible social support [45], which involves daily, systematic and close contact. According to Gallardo-Peralta et al. [10], this practically phenomenological nature of support from a spouse means it is immediately perceived, grasped directly, intuitive and inherent to the spousal relationship. This study progresses the understanding of this association, suggesting that family functioning is a process that can contribute to explaining this invisible nature of support from a partner, which would only play a positive role when perceived as a timely response to psychosocial needs. In contrast to the study hypothesis, social support from a partner is negatively related to social support from relatives in both the general model and the adjusted multigroup model (Figure 2 and Figure 3). One possible explanation lies in the high intensity and interdependence of relationships with close family (spouse and children), leading to a weakening of relationships with other family members. Again, this confirms that the old-age care and support structure in Chile is based on a traditional view of family [6,8]. 

Social support from relatives showed an indirect association with QoL (H3) through family functioning. Intergenerational solidarity-based relationships are central in the aging process in Chile [6]. In this sense, our results show that support from a partner was negatively associated with support from other relatives. It is reasonable to think that in those cases in which an adequate level of social support from the partner is available, the mobilization of social support from other relatives is less relevant [8]. On the other hand, social support from children was associated with an increase in support from other relatives. This result suggests that when mobilizing the support of the adult children, there may be a related increase in support from the adult children’s own networks (daughter-in-law, son-in-law, grandchildren, etc.). In both cases, the support from other relatives is complementary and/or subsidiary to the two main predominant sources (the partner and adult children). In this vein, support from other family members tends to act as a supplement or occasionally as an alternative to support relationships with spouses or children among older adults (SPF theory) [38]. From this perspective, it is unsurprising that social support from relatives is mediated by family functioning rather than there being a direct relationship with QoL. The available empirical evidence offers various examples in this regard; we shall focus on grandchildren and siblings. A study performed by da Silva et al. [75] found that intergenerational grandchild/grandparent relationships were characterized by conflicts in terms of conceptions regarding lifestyle and social and cultural values, but also that intergenerational ties of solidarity, support, affection and care could arise. Moorman and Stokes [76] examined the influences of affinity, contact and functional exchange in the grandparent/adult grandchild relationship upon the depressive symptoms of both members of the dyad. Both studies observed positive and negative effects on the wellbeing of older adults, with the grandchild/grandparent affinity being particularly important. Meanwhile, other research has reported that this family relationship represents a significant social interaction for the older adult, and one that is associated with increased subjective wellbeing and reduced loneliness [77,78]. It should be recalled, too, that the indicator for support in this study also included reciprocity of support, whereby grandparents are also supporting their grandchildren. In a study conducted in Chile, the provision of occasional support to grandchildren was found to be associated with better levels of mental health among grandparents [79]. In the case of siblings, findings have indicated that the long-lasting and strongly intimate nature of this relationship means that it tends to affect wellbeing throughout one’s lifetime, given the high likelihood of not having a partner in old age or of one’s children being busy with their own families. Siblings hence often provide a significant network [49]. In view of all of this, there is a need to continue to conduct analyses that distinguish sources of support, taking into account the high likelihood of intergenerational relationships in old age and their tendency to replace networks made up of partners or children. 

Family functioning was the central variable in our study’s analysis of QoL among older adults. The findings confirmed the existence of a direct relationship (H4). It is also the mediating variable for social support from children, partners and relatives (H1b, H2, and H3). Following Smilkstein [66], family functioning refers to the capacity for adaptation (use of resources to solve crises), partnership (the capacity to share problems and communicate to explore ways of resolving them), growth (the capacity to mature as one moves through the different stages of the family life cycle, facilitating the independence and self-fulfillment of members), affection (the capacity of each family member to experience care and concern and to show emotions such as affection, love, sorrow and anger) and resolve (the capacity to apply the foregoing elements, with each family member devoting time, space and wealth) in the family setting. This involves an in-depth assessment of how families adapt to the new needs of older adults and how they are able to mobilize support in difficult circumstances for older adults [10,37]. In general, the empirical evidence shows that a functional family has positive effects on the wellbeing of older adults [20,50,51]. Following this line of argument, da Silva et al. [75] proposed that happiness among older adults will depend on their establishing harmonious relationships and family unity. It is a process that involves the whole family and which is constructed over the course of time, because straightforward yet difficult actions are required to build a harmonious relationship.

Finally, self-perceived health has a direct association with QoL (H5). Self-perceived health is a reliable and comprehensive measure that replaces other more specific measures of health and disability in the prediction of health outcomes [54,55]. It can be expected to be directly and positively related to QoL, given that it is the most researched domain in QoL [29]. While the general model (Figure 2) confirmed that self-perceived health is related to social support from family members (H5a), this association disappeared in the multigroup model. In this study, we expected to observe a negative association between self-perceived health and social support from family members, based on the assumption that all of the members of the older adult’s network would mobilize in situations involving health problems, including networks made up of other family members. However, this contrasts with the fact that the older adults are generally in worse health, and the numbers of people in their networks are hence increasingly limited, whether due to death or difficulties with mobility [80]. It might therefore be expected that people in worse health would have a lower perception of the availability of social support, which was how social support was measured in this study. In fact, several studies have not found an association between the intensity of the grandparent/grandchild relationship and the health of the grandparent [81,82].

It should be noted that although this study had a multi-ethnic sample, no specific comparative analyses were performed by ethnic group. It is important to note that the sample is constituted by a population living in predominantly rural settings, so that older people face common challenges that directly affect the composition of support networks. We are mainly referring to the depopulation of rural areas and the changes that this process has in family structures [8,57]. The movements of the younger population to predominantly urban environments and the narrowing of families are at the origin of a process according to which non-family social networks are becoming increasingly important, whether neighbors, friends, or members of the social groups in which they participate [9]. Moreover, gender differences constitute a particularly important element of study, given the important sexual division of social roles in these environments. Therefore, we find a context reluctant to generate changes in the traditional organization of care, to some extent regardless of ethnicity. 

In summary, the findings of this study confirm that Chile maintains a traditional structure of care and well-being for older people, i.e., the family remains the main support network related to QoL. Specifically, the results showed that only the social support of children is directly and positively related to QoL, while the social support of partners and other family members is indirectly related, through family functioning, to QoL.

### 4.1. Implications for Practice 

Gerontological social intervention, especially in Latin America, focuses on assessing the support or help of family members as a whole. In this study we have seen the differentiated role of the sources of family support: from a partner, children and other relatives. This reinforces the need to include in the assessment (diagnosis) a specific analysis of the structure of the household and the structure of social support networks among older people, in order to design more complex social interventions in the gerontological field that take into account the heterogeneity of family networks.

Along with these findings, this study also shows that the quality of family relationships is more important than the quantity of social relationships. Many social interventions focus on increasing social relationships among older people, either to avoid loneliness or social isolation, but it is necessary to return to interventions aimed at improving or strengthening existing family ties. In Chile, social policies such as the Programa Vínculos (“Ties Program”) focus on the prevention of exclusion and social isolation among older adults by increasing the density of their social and community support network. This study investigates the role played by family sources of social support and its impact on family functioning. Our results suggest that support from different sources within the family is specifically associated with both family functioning and quality of life. In other words, these differences in the pattern of association of social support from different sources implies relevant knowledge to fine-tune the design of policies and programs to combat loneliness and social isolation and to improve quality of life. In addition, our results suggest that the range of relevance variables is different between men and women, which also implies relevant knowledge for the design of programs such as the one mentioned above.

Finally, this study shows the relevance of intergenerational relationships in old age; this point should also be strengthened in gerontological social intervention. As we have advanced in this study, it is increasingly likely that we will grow old with our siblings (being the most stable social networks in our lives) and that we will spend a large part of our old age together with grandchildren. 

### 4.2. Limitations of the Study

Various limitations of this study should be taken into account. First, the study had a transversal design, meaning that causal relationships with QoL could not be established and we were restricted to analyzing variables that were associated with QoL. Second, the sample was not representative of older adults in Chile. Although valid inferences could be drawn due to quota sampling, representativeness is not guaranteed owing to the lack of random selection. In any case, the sample would be representative of the rural areas in the regions of Arica, Parinacota and Araucanía, rather than being a national sample. In relation to the characteristics of the sample, it is important to analyze the possible bias that older people in rural environments have in making an excessively positive assessment of their well-being. It is also important to bear in mind, in order not to generalize the findings to the ageing population in Chile, that the profile of the sample was typical of older people living in rural areas, i.e., with a low level of education, high illiteracy, most of them having a partner and having a low socio-economic level. Third, this study assessed social support from certain sources (spouse, children, and relatives), but the evaluation of family functioning was carried out on a global family basis. In future research, it would be useful to include an evaluation of family functioning that is differentiated for each family member.

## 5. Conclusions

Following SPF theory, older adults tend to mobilize their social support networks in response to difficult situations, such as illnesses and other needs. In this study, we confirmed that social support from children, a partner and family members is indirectly associated with QoL through family functioning. This means it cannot be taken for granted that the mere existence of a support network is related to QoL; rather, older adults need to perceive that this source of social support is functional in terms of being able to satisfy their physical and social needs. In view of this, social policies need to encourage interventions that focus on the social integration of older adults but also work with their family environment to ensure they are provided with social support that is adequate for their needs in terms of emotion, information and advice.

## Figures and Tables

**Figure 1 ijerph-19-09247-f001:**
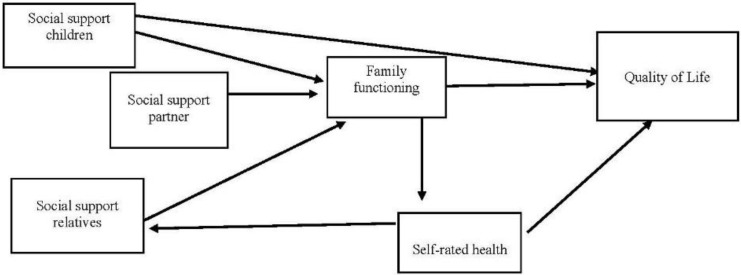
Hypothesized model.

**Figure 2 ijerph-19-09247-f002:**
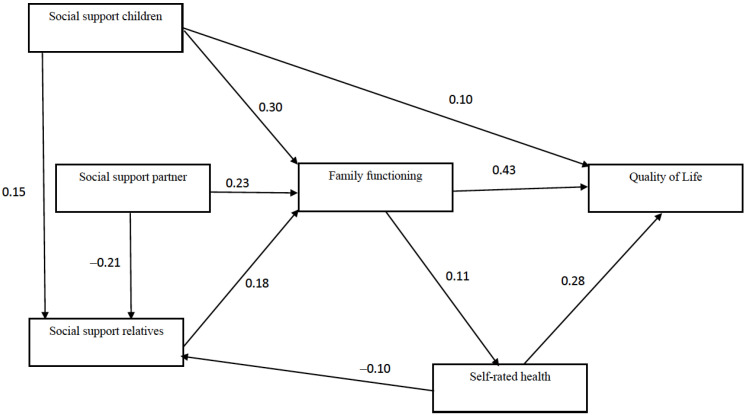
Model fitted for the whole sample. Standardized coefficients.

**Figure 3 ijerph-19-09247-f003:**
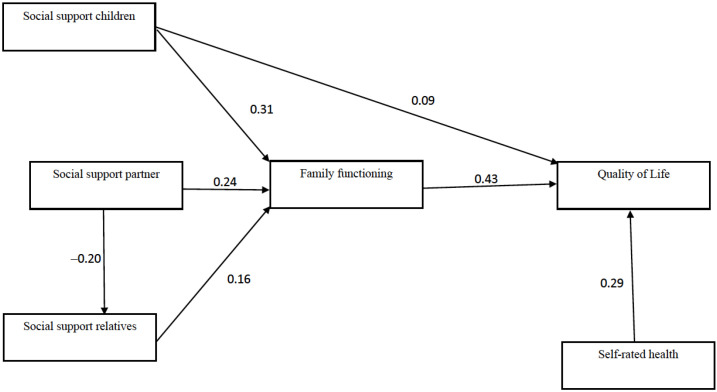
Multi-group analysis (men and women). Standardized coefficients.

**Table 1 ijerph-19-09247-t001:** Participants’ characteristics.

Variable	Categories	*n* (%)
Gender	Women	393 (49%)
	Men	407 (51%)
Age groups	60–69 years	341 (43%)
	70–79 years	311 (39%)
	80+ years	148 (18%)
Marital status	Married or cohabiting	434 (54%)
	Single	124 (15%)
	Widow	190 (24%)
	Divorced, separated	52 (7%)
Household structure	Single person	197 (25%)
	Couple without children	264 (33%
	Couple with children	132 (16%)
	Only with children	108 (13%)
	Other family members	96 (12%)
	Non-relatives	3 (1%)
Household size	1 person	197 (25%)
	2 persons	332 (42%)
	3 persons	95 (12%)
	4 persons	77 (10%)
	5 or more persons	99 (11%)
Residence	North (region of Arica y Parinacota)	311 (39%)
	South (region of The Araucanía)	489 (61%)
Education	Primary School incomplete	433 (54%)
	Primary School	245 (31%)
	High School or vocational education	108 (13%)
	Higher education	14 (2%)
Ethnicity	Indigenous	569 (71%)
	Non-indigenous	231 (29%)

**Table 2 ijerph-19-09247-t002:** Descriptive statistics and correlations *(Pearson’s r)* for the main variables of the study.

	Mean	SD	1	2	3	4	5
1. Social support from a partner	3.11	3.52					
2. Social support from children	4.31	3.28	0.016				
3. Social support from relatives	2.50	3.18	−0.210 **	0.134 **			
4. Family functioning	7.06	3.12	0.202 **	0.327 **	0.159 **		
5. QoL	81.27	13.99	0.136 **	0.268 **	0.127 **	0.488 **	
6. Self-rated health	2.29	0.73	−0.005	0.085 *	−0.066	0.092 **	0.332 **

SD: standard deviation. ** *p* < 0.01; * *p* < 0.05.

**Table 3 ijerph-19-09247-t003:** Multi-group analysis, model comparisons (testing invariance).

	df	CMIN (χ^2^)	*p*	NFI	IFI	RFI	NNFI
Structural weights	7	6.298	0.505	0.011	0.011	−0.043	−0.045
Structural residuals	6	13.364	0.038	0.024	0.025	−0.006	−0.006

CMIN: Chi-square; df: degrees of freedom. NFI: Normed Fit Index; IFI: Incremental Fit Index; RFI: Relative Fit Index; NNFI: Non-Normed Fit Index (differences between models).

## Data Availability

Not applicable.

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
