# Peer review of "Aging and Family Relationships among Aymara, Mapuche and Non-Indigenous People: Exploring How Social Support, Family Functioning, and Self-Perceived Health Are Related to Quality of Life"

_ijerph, 2022, doi:10.3390/ijerph19159247_

Round 1
Reviewer 1 Report
This is a courageous piece of work trying to relate several complex concepts that have even more complex relationships among them. The paper is well written. While I’m generally convinced by the measurement models, I am more skeptical about the modelling of the relationships between the various concepts. My major concern is that everything is endogenous, so no causal relationships are identified (as the authors note in the discussion). This severely hinders the study, in that no policy recommendations can really be drawn. Besides, even if a causal impact of family functioning on QoL could be established, it is difficult to see how family functioning is amenable to policy intervention.
Other points:
1. The direction(s) of the relationship(s) between self-rated health and QoL is missing in figure 1 (perhaps a formatting issue).
2. I understand that the model needs to be identified and it is not possible to include all potential relationships between the concepts; however I find it strange that social support from children directly “impacts” QoL, but support from spouses and other relatives can only “impact” QoL through family functioning. Similarly, why health would “impact” support from other relatives but not from children or spouses is strange. There are also complex substitute/complementary relationships between different sources of support that are not accounted for.
3. On p. 4, line 164 (H3), I think the authors meant to write “positively and indirectly”, not “directly and indirectly”.
4. P. 8, line 321, it is unclear what is the “unbiased estimation of confidence intervals method”.
5. Do the authors see a potential role for non-relative (friends, neighbors) sources of support in the context of Chile? In many countries this is a substantial source of support.
6. In the data, how did the authors distinguish between, for example, not having children (or not having other relatives) and having children (or other relatives) but not receiving support from them? I.e., two different types of zeroes.
Author Response
The authors wish to thank both the reviewers and the editor for their contributions. The comments and suggestions provided have made a decisive impact in helping us to produce a (hopefully) improved new manuscript. The authors’ view is that the new manuscript has been significantly altered. Pressures of space necessarily lead to taking decisions at the time of finalising the manuscript. The authors have made efforts to take the suggestions into account, within the framework of the content of the editor’s decision. In any event, and along the same lines, we hope to receive new comments and suggestions to help us address any remaining elements that it is considered could be improved.
We used red color to highlight the changes in the manuscript.
REVIEWER 1
This is a courageous piece of work trying to relate several complex concepts that have even more complex relationships among them. The paper is well written. While I’m generally convinced by the measurement models, I am more skeptical about the modelling of the relationships between the various concepts. My major concern is that everything is endogenous, so no causal relationships are identified (as the authors note in the discussion). This severely hinders the study, in that no policy recommendations can really be drawn. Besides, even if a causal impact of family functioning on QoL could be established, it is difficult to see how family functioning is amenable to policy intervention.
RESPONSE. Thank you very much for your interesting insights. Indeed, the weakness of the study lies in its cross-sectional design, which does not allow causal relationships to be established and only shows the associations observed. In Chile there are only a few longitudinal studies using a general population sample, but there is no specific research on older people with a longitudinal design, and perhaps the most complex aspect is the lack of research that includes indigenous and non-indigenous older people among its participants. people. Perhaps that is the main strength of this study. From our point of view, the results obtained are useful for policy makers, and specifically for the design or improvement of programs whose objective is to improve the quality of life of the elderly and reduce social isolation. To show this potential, we have included a paragraph on rows 580-592. In general, we consider that the improvements introduced in the manuscript based on the suggestions and observations of the reviewers have increased the clarity of the text and the implications of the results obtained in our research.
Other points:
- The direction(s) of the relationship(s) between self-rated health and QoL is missing in figure 1 (perhaps a formatting issue).
DONE. There is indeed a formatting error and we have replaced the figure so that it is clearer now.
- I understand that the model needs to be identified and it is not possible to include all potential relationships between the concepts; however I find it strange that social support from children directly “impacts” QoL, but support from spouses and other relatives can only “impact” QoL through family functioning. Similarly, why health would “impact” support from other relatives but not from children or spouses is strange. There are also complex substitute/complementary relationships between different sources of support that are not accounted for.
RESPONSE. Thank you very much for your reflection. The authors believe that the hypotheses that make up Figure 1 are adequately described and supported by empirical evidence. In fact, we devote pages 3 to 5 to a detailed description of the empirical basis for these hypotheses. When the hypotheses were not supported by the results obtained, we proceeded to a discussion of the hypotheses. After re-examining the text in the light of the reviewer's comments, we have added an argument that strengthens the arguments concerning the association between the different sources of support (especially in the case of "social support from relatives") (row 489).
- On p. 4, line 164 (H3), I think the authors meant to write “positively and indirectly”, not “directly and indirectly”.
DONE. Thank you.
- P. 8, line 321, it is unclear what is the “unbiased estimation of confidence intervals method”.
DONE It was a wording or writing error. It has been amended
- Do the authors see a potential role for non-relative (friends, neighbors) sources of support in the context of Chile? In many countries this is a substantial source of support.
DONE. Although this is briefly proposed in the introduction (row 38-39) it is incorporated at the end of the discussion (rows 545-558)
- In the data, how did the authors distinguish between, for example, not having children (or not having other relatives) and having children (or other relatives) but not receiving support from them? I.e., two different types of zeroes.
RESPONSE The questionnaire had a filter question for these categories, i.e. do you have a partner? do you have children? do you have grandchildren? etc. Thus, when asked "who do you turn to on a day-to-day basis to solve practical problems or emotional needs (social support), those who did not have a support network were automatically included as a response category "does not apply because you do not have the network" (category=0) and those who had the network but perceived that these sources of support - spouse, children, etc. - did not provide the support responded "never" (category =1). We have included this explanation in the description of the perceived social support questionnaire.
Reviewer 2 Report
Dear Authors,
Thank you for an interesting work. All the details of the review are provided with attached file.
Best regards

Author Response
The authors wish to thank both the reviewers and the editor for their contributions. The comments and suggestions provided have made a decisive impact in helping us to produce a (hopefully) improved new manuscript. The authors’ view is that the new manuscript has been significantly altered. Pressures of space necessarily lead to taking decisions at the time of finalising the manuscript. The authors have made efforts to take the suggestions into account, within the framework of the content of the editor’s decision. In any event, and along the same lines, we hope to receive new comments and suggestions to help us address any remaining elements that it is considered could be improved.
We used red color to highlight the changes in the manuscript.
Dear Authors, Thank you for the possibility to learn about your research in the field of QOL among older adults. The manuscript is relatively well written, with visible and clear structure, and have adequate references. Some general comments. The title of the work could reflect the particular value of researching two ethnic minorities of the country (or the areas they live in). The Introduction part, although could be more concisely written, is reader friendly for having each hypothesis separately followed by relevant theoretical and empirical justification. Already at this stage a couple of sentences with references is necessary to introduce a description of the specifics of the Ayamara and Mapuche population (in terms of the family functioning and health). The Method section includes a valuable information about the preparation to the research. However, some parts need amendment, with provision of full inclusion and exclusion criteria being the most important. Also an explanation of data analysis for participants without a partner or children could help better understand the given data (e.i. Tab. 2). Description of the Results has to be supplemented with the missing table 3, without which it is not possible to make a proper review. Discussion section includes interesting remarques about the role of reciprocity, yet this was not the subject of the study nor a separate part of the analysis and therefore these reflections need to be shortened. Limitations of the study could include the role a participant personality might play in biasing the results (e.g. optimistic or melancholic characteristics may cause the occurrence of apparent associations between self-rated health and QOL). Moreover, the specific profile of the sample should be considered (low education level 85% have primary education or less, and low rate of separated/divorced participants 7% ) as well as the economic status of the studied population, if known. On one hand, that could help a valid generalisation of the results, on the other might be pointed out as a unique strength due to approaching a population that is underserved and difficult to reach.
RESPONSE. Following the reviewer's recommendation, we have amended the title. Given the length of the introduction, we have included a short paragraph to contextualise the social/family organisation of indigenous older people and to give some data on their health. The analysis of reciprocity has been summarized/shortened. The suggestions in the limitations of the study have also been incorporated, as well as the missing table 3.
Detailed comments throughout the text by given parts or lines:
- Footnote on the title page provides year 2021 edition error?
DONE. Edited
- 5-8 authors names are not assigned to their universities
DONE
- 23 a typo with big letters for gerontological social intervention - Is reference style in accordance to authors guidelines
DONE
- 36-37 this might be seen in parallel as a part of traditional multigeneration culture
DONE. We have included this proposal, including a Chilean bibliographic reference.
- 48-50 the meaning of the sentence is not clear
DONE. We have changed the wording to make the idea clearer.
- 53 QOL being - 56 listed are rather phenomena or factors than processes
DONE. There was indeed a mistake in the terms and we have incorporated a proposal to make the difference between process, determinants, etc.
- 105-107 the meaning of the sentence is not clear
DONE. The argumentation has been modified to give clarity to the Chilean protection system based on a subsidiary state.
- 117-121 due to its length the sentence is difficult to understand
DONE
- 122 the statement has to be softened as parts of the model miss empirical support, so please mention that this is a hypothetical model or Authors assumption
DONE. The paragraph has been reworded to increase clarity and to show that this is an assumption/hypothesis of the authors (lines 121-127).- Fig. 1 two of the associations have an unindicated direction
- Fig 1 nad Fig 2 formatting marks are visible
DONE. There is indeed a formatting error and we have replaced the figure so that it is clearer now.
- 172 frequency of contact?s
DONE. We have re-checked the study and corrected it to show the findings directly found.
- 234 -off point would be helpful - Recruitment all eligibility criteria should be listed
DONE. We have included the inclusion criteria " Inclusion criteria were age (60 years or older), residence (rural setting), health status (no severe cognitive impairment) and voluntary participation in the study".
- 256-7 the meaning of the sentence is not clear
DONE. We have expanded the sentence to better understand the procedure regarding the place where the interview was conducted.
- 267-268 XXX should be replaced
DONE.
- 302 is resolve used as a noun?/possibly a solution could fit?/ not clear
DONE. we have made a change in the wording to provide a better understanding on how the scale is calculated and interpreted.
- 315 looks like an inconsistency: scale range one to five or 0 to 4?
DONE. We have made a mistake in the approach, we have included the way in which it was evaluated.
- 324-236 and 384 the NFI abbreviation is not introduced
DONE
- Table 2 complementing the correlation matrix with self-rated health would be interesting
DONE
- 363 the manuscript does not contain Table 3 due to this reason proper review of large parts of discussion is imposible.
DONE. We regret the omission. Now table 3 is included in the manuscript
- 556-560 long sentence; it fits better to the discussion part
DONE.
Reviewer 3 Report
I very much enjoyed reading this detailed and interesting paper that contributes in a useful way to the wider literature about family care for older people in Chile. The paper is well written, the methods are appropriately described and the findings and discussion are detailed and well presented.
There were just a few areas where some work could be done to add clarity to the paper.
The paper is about families in Chile, a majority world country and, as noted by the authors, older people are more reliant on families for care and support in these regions. In the background section it would be helpful to delineate between background research that has been conducted in similar regions and research that has been conducted in, for example, the global north. The location of relevant research is noted for some papers but this could be more consistent.
In the findings section I was unable to locate Table 3.
Further, in this section, it would have been helpful to provide more detailed discussion and description of Figure 3 as this is the crux of the paper. For readers without a background in statistical analysis it would be helpful to have more clarity in how Figure 3 is described and explained.
The discussion section is fairly long and complex and a short summary at the end of this section to highlight key findings would be helpful.
One of the limitations noted in the paper is the particular sample that was included in the research with a high proportion of people from indigenous communities. Rather than presenting this simply as a limitation it would have been interesting to say more about the particular family structures among the indigenous communities and what this research tells us about the experiences of older people in these communities.
Overall the paper was dense and covered a lot of ground. One suggestion might be to reduce the number of hypotheses discussed to focus on those that reveal new knowledge.
Author Response
The authors wish to thank both the reviewers and the editor for their contributions. The comments and suggestions provided have made a decisive impact in helping us to produce a (hopefully) improved new manuscript. The authors’ view is that the new manuscript has been significantly altered. Pressures of space necessarily lead to taking decisions at the time of finalising the manuscript. The authors have made efforts to take the suggestions into account, within the framework of the content of the editor’s decision. In any event, and along the same lines, we hope to receive new comments and suggestions to help us address any remaining elements that it is considered could be improved.
We used red color to highlight the changes in the manuscript.
I very much enjoyed reading this detailed and interesting paper that contributes in a useful way to the wider literature about family care for older people in Chile. The paper is well written, the methods are appropriately described and the findings and discussion are detailed and well presented.
There were just a few areas where some work could be done to add clarity to the paper.
The paper is about families in Chile, a majority world country and, as noted by the authors, older people are more reliant on families for care and support in these regions. In the background section it would be helpful to delineate between background research that has been conducted in similar regions and research that has been conducted in, for example, the global north. The location of relevant research is noted for some papers but this could be more consistent.
RESPONSE. We thank you for your suggestion, we have incorporated the country where the research was conducted in most of the studies. The reviewers have warned that the search of previous articles and pieces of research has been ambitious. In this sense, we tried to reach studies conducted in Chile and Latin America, but when it was not possible to find evidence to support the hypotheses we included studies from various regions.
In the findings section I was unable to locate Table 3.
DONE. We regret the omission. Now table 3 is included in the manuscript
Further, in this section, it would have been helpful to provide more detailed discussion and description of Figure 3 as this is the crux of the paper. For readers without a background in statistical analysis it would be helpful to have more clarity in how Figure 3 is described and explained.
DONE
The discussion section is fairly long and complex and a short summary at the end of this section to highlight key findings would be helpful.
DONE.
One of the limitations noted in the paper is the particular sample that was included in the research with a high proportion of people from indigenous communities. Rather than presenting this simply as a limitation it would have been interesting to say more about the particular family structures among the indigenous communities and what this research tells us about the experiences of older people in these communities.
DONE. Thank you very much for this comment, it has been discussed by the authors and we have tried to clarify this issue in the discussion (rows 558-570).
Overall the paper was dense and covered a lot of ground. One suggestion might be to reduce the number of hypotheses discussed to focus on those that reveal new knowledge.
RESPONSE. Thank you very much for this comment, which has given rise to an interesting discussion among the authors. We have tried to emphasize the most relevant elements derived from the hypotheses and the results obtained to contrast these hypotheses. Thus, for example, we have reduced the discussion on reciprocity, but we have slightly increased the space dedicated to the association found between the three sources of family support considered (this last aspect was not included in the hypotheses, but the results focused on it as an element of relevance). Finally, we have emphasized the relevance of the results for the practice and design of intervention programs.
Round 2
Reviewer 1 Report
I wish to thank the authors for their work and their revision. I have no further comments.
Author Response
The authors would like to thank you for this second round of review of our manuscript. We appreciate the time spent in peer review of our article (we do peer review ourselves and are aware of the work involved). Best regards.
Reviewer 2 Report
Dear Authors,
Thank you for considering all comments. The manuscript is now significantly improved in my opinion. The remaining remarques are described in the attached file. All the best for you further research!

Author Response
The authors would like to thank you for this second round of review of our manuscript. We appreciate the time spent in peer review of our article (we do peer review ourselves and are aware of the work involved).
Thank you very much for bringing to our attention the remaining areas for improvement. We have made all changes, with one exception. After an interesting discussion among the authors, we have decided to keep the title (not to include the reference to Chile). We have made this decision on the basis of two criteria. First, the reference to Chile appears both in the abstract and early in the text. Second, the reference to the Mapuche people (and also to the Aymara people, although to a lesser extent) immediately situates the study in Chile, so the reference to the country in the title would even be redundant. Thus, the title focuses more on the multiethnic dimension of the sample (as suggested by the reviewers in the first round) than on a specific country. In any case, if deemed necessary, we are open to reconsider this issue.